# The Development of the Genic SSR Markers for Analysis of Genetic Diversity in Gooseberry Cultivars

**Elena O. Vidyagina [1], Vadim G. Lebedev [1], Natalya M. Subbotina [1], Ella I. Treschevskaya [2], Tatyana N. Lebedeva [3] and Konstantin A. Shestibratov [1,2,*]**

[1] Branch of the Shemyakin-Ovchinnikov Institute of Bioorganic Chemistry of the Russian Academy of Sciences, Science Avenue 6, 142290 Pushchino, Russia; vidjagina@mail.ru (E.O.V.); vglebedev@mail.ru (V.G.L.); natysubbotina@rambler.ru (N.M.S.)

[2] Faculty of Forestry, Voronezh State University of Forestry and Technologies Named after G.F. Morozov, 8 Timiryazeva Str., 394613 Voronezh, Russia; ehllt@yandex.ru

[3] Institute of Physicochemical and Biological Problems of Soil Science, Russian Academy of Sciences, Institutskaya Str. 2, 142290 Pushchino, Russia; tanyaniko@mail.ru

[*] Correspondence: schestibratov@bibch.ru or schestibratov.k@yandex.ru; Tel.: +7-496-7330-966

**Abstract:** Gooseberry is one of the most promising and underrated small fruit crops. There is a need to improve and genotype the existing cultivars, and this work requires the use of new efficient DNA marking techniques. Marker-assisted selection (MAS) is a modern approach for work with genetic resources. However, the genetic determinants of key qualitative traits are poorly studied. Therefore, we developed SSR markers located in flavonoid biosynthesis genes that can affect the resistance of plants to biotic and abiotic stresses to study the genetic diversity of gooseberry cultivars grown in the Russian Federation and varying in genetic and geographical origin. We have genotyped 22 gooseberry cultivars using a set of 19 of our original microsatellite markers and two neutral previously published ones. A total of 53 alleles were found. Nine of the 21 analyzed loci were polymorphic. The most polymorphic loci of flavonoid biosynthesis were found in the *DFR* gene (RucDFR1-2, RucDFR1-3, RucDFR2-1), their mean PIC (polymorphism information content) being 0.65, comparable to the PIC values of neutral markers. Our set of gene-targeted SSR markers showed that most of all the studied gooseberry cultivars differed in origin, based on which they were divided into three groups: European cultivars, Russian cultivars, and interspecies hybrids. Thus, the polymorphic markers can be used for cultivar identification and certification as well as for the marker-assisted selection of gooseberry plants having different origins and phenotypic traits.

**Keywords:** genetic variability; marker-assisted selection; microsatellites; gooseberry; *Ribes uva-crispa*

## 1. Introduction

Gooseberry is an important berry crop: it has a high nutritional value and can be a source of vitamins, minerals, and other biologically active substances [1–3]. The berries are attractive in color and taste and can tolerate long transportation. These properties make gooseberry very popular in the countries of Northern and Eastern Europe, as well as in the Baltic States [4]. Gooseberry has a long cultivation history in Russia: it has been mentioned in monastic chronicles since the 11th century, whereas in Europe since the 15th century [5]. The Russian Federation is among the countries with the largest gooseberry production volumes [6]. This is not only owing to the use of the latest agricultural technologies but also due to the introduction of new productive and sustainable cultivars. Gooseberry breeding, previously aimed at improving qualitative and quantitative traits of berries, has recently refocused on the development of pest- and disease-resistant cultivars [4] and obtaining desired chemical composition of berries [7,8].

More and more studies are now investigating gene linkage for economically valuable traits [9]. For the genus *Ribes*, strategies are being developed that use molecular markers

for breeding cultivars possessing desired traits [4]. Such strategies can speed up the early stages of breeding [10]. Marker-assisted selection (MAS) is seen as the main approach for obtaining new plant cultivars in the 21st century [11]. Marker sequences can also be applied to identify existing cultivars and establish their origin much more efficiently and at any growth stage [9,12]. Successful use of marker sequences requires having genetic maps that link markers to phenotypic or biochemical traits. Such maps can be used to identify the quantitative traits loci (QTLs), such as phenological and phenotypic traits (agronomic and qualitative traits of fruits). The EU project "Core collection of Northern European gene pool of *Ribes*" (RIBESCO, www.ribes-rubus.gf.vu.lt, accessed on 1 May 2021) was aimed at more efficient breeding of *Ribes* and preservation of its gene pool. Molecular analyses based on SSR markers were performed for 800 cultivars of the genus *Ribes*. This approach was shown applicable for cultivar identification and certification, as well as for tracing links between genetic and phenotypic components [13]. In that project and subsequent studies, molecular marking was mainly focused on currant cultivars; hence there are incomparably fewer data on gooseberry cultivars.

The growing attention to the genus *Ribes* is due to the proven benefits of eating their fruit. The berries are rich in polyphenolic compounds, particularly anthocyanins and flavonols [4,14,15]. To obtain new cultivars of gooseberries, breeding used to go along the path of increasing the size of the fruits, changing their organoleptic properties, and also increasing the yield. At the moment, changes in climatic and conditions and general globalization have led to the fact that there is a tendency to obtain cultivars resistant to the action of biotic and abiotic environmental factors [16]. Gooseberry cultivars grown in the Russian Federation, due to the peculiarities of the climate, are distinguished by their resistance to biotic stress factors and a high level of winter hardiness [17]. According to the RHS (Royal Horticultural Society) hardiness rating in the Russian Federation, gooseberry cultivars with a rating from H5 to a maximum of H7 are mainly grown [18]. In this regard, of particular interest to study the features of synthesis of polyphenols in different cultivars of gooseberries. For plants, these agents play an important role since they not only attract pollinators of flowers and animals that feed on fruits but also increase resistance to the influence of biotic and abiotic factors [19]. The expression of flavonoid genes changes in response to low temperatures [20]. Therefore, developing *Ribes* cultivars having certain features of biosynthesis of anthocyanins and flavones is an important objective for many breeders [21]. Flavones and anthocyanins are synthesized through the flavonoid synthesis pathway. Its stages, enzymes, and regulators are well known [22–24]. Abreu et al. (2020) mapped the QTLs of polyphenolic metabolites, particularly anthocyanins and flavonols, in blackcurrant (*Ribes nigrum* L.) [9]. Not long ago, some specific features of the flavonoid pathway genes in *Ribes,* including gooseberries, were described [25]. However, variations in these genes within the same species in different cultivars were not determined. The regulatory genes of the flavonoid pathway were shown to be as important as the structural ones. The syntheses of anthocyanins, proanthocyanins, and flavonols are regulated by the transcriptional complex MYB-bHLH-WD40 (MBW). The complex includes three classes of regulatory proteins (R2R3-MYB, bHLH, and WD40) that control the late stages of the flavonoid pathway [26]. The same regulatory proteins play a key role in determining the cell fate of trichomes during spine formation. These genes are used as marker genes to establish the relationship between the genetic component and the degree of thorniness in various plants [27].

Simple sequence repeats (SSRs) are one of the DNA-based markers used for establishing genetic profiles of cultivated plants, investigating genetic relationships among plants, and evaluating genetic diversity [12,28]. The high information content of SSRs can be used for QTL mapping [12,29]. The development and use of SSRs provide clear advantages in the outbreeding of diploid species, such as *Ribes* spp. [12]. From the molecular perspective, gooseberry still remains a poorly studied crop when compared to other representatives of the genus *Ribes*. At the same time, the value of gooseberry as an industrial berry crop is being revised [4,30,31]. The increasing interest in this crop requires creating qualitatively

new cultivars. There are several thousand gooseberry cultivars [30], but they are poorly characterized genetically. Some studies assessed the genetic diversity of some gooseberry cultivars using random genomic SSR markers [12,32,33]. Yet, no genetic diversity studies used markers located in genes responsible for any economically valuable traits. Our study has evaluated the genetic diversity of popular gooseberry cultivars grown in the Russian Federation from different breeding centers with different economically valuable traits. We developed SSR markers based on the nucleotide sequences of structural and regulatory genes of the flavonoid biosynthesis pathway in *Ribes uva-crispa* L., available in the GenBank database of the National Center for Biotechnology Information (NCBI). The markers were genotyped in 22 gooseberry cultivars bred in Russia, Great Britain, Latvia, and Finland.

## 2. Materials and Methods

### 2.1. Plant Materials

We used 22 gooseberry cultivars (Table 1) to genotype SSR loci of flavonoid biosynthesis genes. The cultivars were kindly provided by Microklon (Pushchino, Russia) and the N.V. Tsitsin Main Botanical Garden of the Russian Academy of Sciences (Moscow, Russia). The selected cultivars are popular in the Russian Federation and differ in geographic and genetic origins. In addition, the studied cultivars are characteristic representatives of different breeding centers, differ from each other in varying degrees of temperature tolerance, prickliness, and color of berries. All studied cultivars are resistant to the action of various biotic factors. A pedigree reconstruction was constructed in order to analyze the inheritance of traits and loci of interest in the gooseberry cultivars under study (Figure 1).

### 2.2. Search for Microsatellite Loci and Primer Design for PCR

SSR loci were developed based on the sequences of *Ribes uva-crispa* flavonoid biosynthesis genes from the NCBI database (https://www.ncbi.nlm.nih.gov/, accessed on 1 May 2021). Loci RJL-2 and RJL-6 were taken from the published study [34]. Locus *RnMYB-1* was developed based on the *MYB* nucleotide sequence of 10 *Ribes nigrum* plants of the same genus *Ribes* (LN736314). The loci LfMYB-1 and LfMYB-2 were developed based on the known sequence of the gene *MYB113* from *Liquidambar formosana*, a representative of the same order *Saxifragales* (KU987934). WebSat software was used to detect microsatellite loci in the gene sequences [35]. Primers were designed using the program Primer 3 (http://primer3.org/, accessed on 1 May 2021). The main criteria for primer design were as follows: primer length of 18–27 bp (the optimal length was 21–22 bp), 40–80% GC content; primer annealing temperature of about 60 °C; expected amplicon length of 150–400 bp. Primers were synthesized by Synthol (Syntol Comp., Moscow, Russia). Table 2 presents primers used in this study.

**Table 1.** Gooseberry cultivars genotyped in the study.

| Cultivar | Abbr. | Genetic Origin and Background | Degree of Spinosity | RHS Hardiness Rating | Fruit Color | Origin |
|---|---|---|---|---|---|---|
| Avenarius | Ave | - | Medium | H7 | red | Russia |
| Angliyskiy Zeleniy | AnZ | - | High | H7 | yellow-green | Russia |
| Chorny Negus | ChN | Anibut × R.cuccirubra | High | H7 | black | Russia (Michurinsk) |
| Chernomor | Che | 2152 × mixed pollen (Phenicia, Zeleniy Butylochniy, Brazilskiy, Seyanets Maurera) | Lower | H7 | black | Russia (Michurinsk) |
| Consul (Senator) | Con | Chelyabinskiy Zeleniy × Afrikanets | Lower | H6 | dark red | Russia (Southern Urals) |
| Chelyabinskiy Zeleniy | ChZ | Houghton × Angliyskiy Zholtiy | Medium | H7 | green | Russia (Southern Urals) |
| Grushenka | Gru | Severniy Kapitan × Moskovskiy Krasniy × GF 595-33 | Lower | H7 | nearly black | Russia (Moscow) |
| Invicta | Inv | Keepsake × (Whinham's Industry ×Resistentia) | Medium | H7 | pale green | England, U.K. |
| Kolobok | Kol | Rozoviy-2 × Smena | Lower | H5 | dark red | Russia (Moscow) |
| Kursu Dzintars | KuD | Stern Razhig × Pellervo | High | H7 | yellow | Latvia |
| Laskoviy | Las | Neslukhovskiy × Kolobok | Lower | H7 | red | Russia (St. Peterburg) |
| Lepaan Red | LeR | - | Medium | H7 | dark red | Finland |
| Malahit | Mal | Chorny Negus × Phenicia | Lower | H7 | bright green | Russia (Michurinsk) |
| Moskovskiy Krasniy | MoK | Avenarius (open pollination) | Lower | H6 | dark red | Russia (Moscow) |
| Nezhniy | Nez | - | Lower | H6 | pale green | Russia |
| Rozoviy-2 | Ro2 | Seyanets Lefora×Phenicia | Medium | H6 | dark red | Russia (Moscow) |
| Russkiy | Rus | Careless× mixed pollen (Houghton, Oregon, Karri, Shtamboviy) | Medium | H6 | dark red | Russia (Michurinsk) |
| Russkiy Zholtiy | RuZ | Bud mutation of Russkiy | Medium | H6 | yellow | Russia (Michurinsk) |
| Severniy Kapitan | SeK | 310-24 × Rozoviy-2 | Lower | H7 | nearly black | Russia (Moscow) |
| Seyanets Lefora | SeL | Eduard Lefor (open pollination) | Medium | H7 | red | Russia (Vologda Region) |
| Triumfalniy | Tri | - | Medium | H7 | yellow | Russia |
| Vladil | Vla | Chelyabinskiy Zeleniy × Afrikanets | Lower | H6 | red | Russia (Southern Urals) |



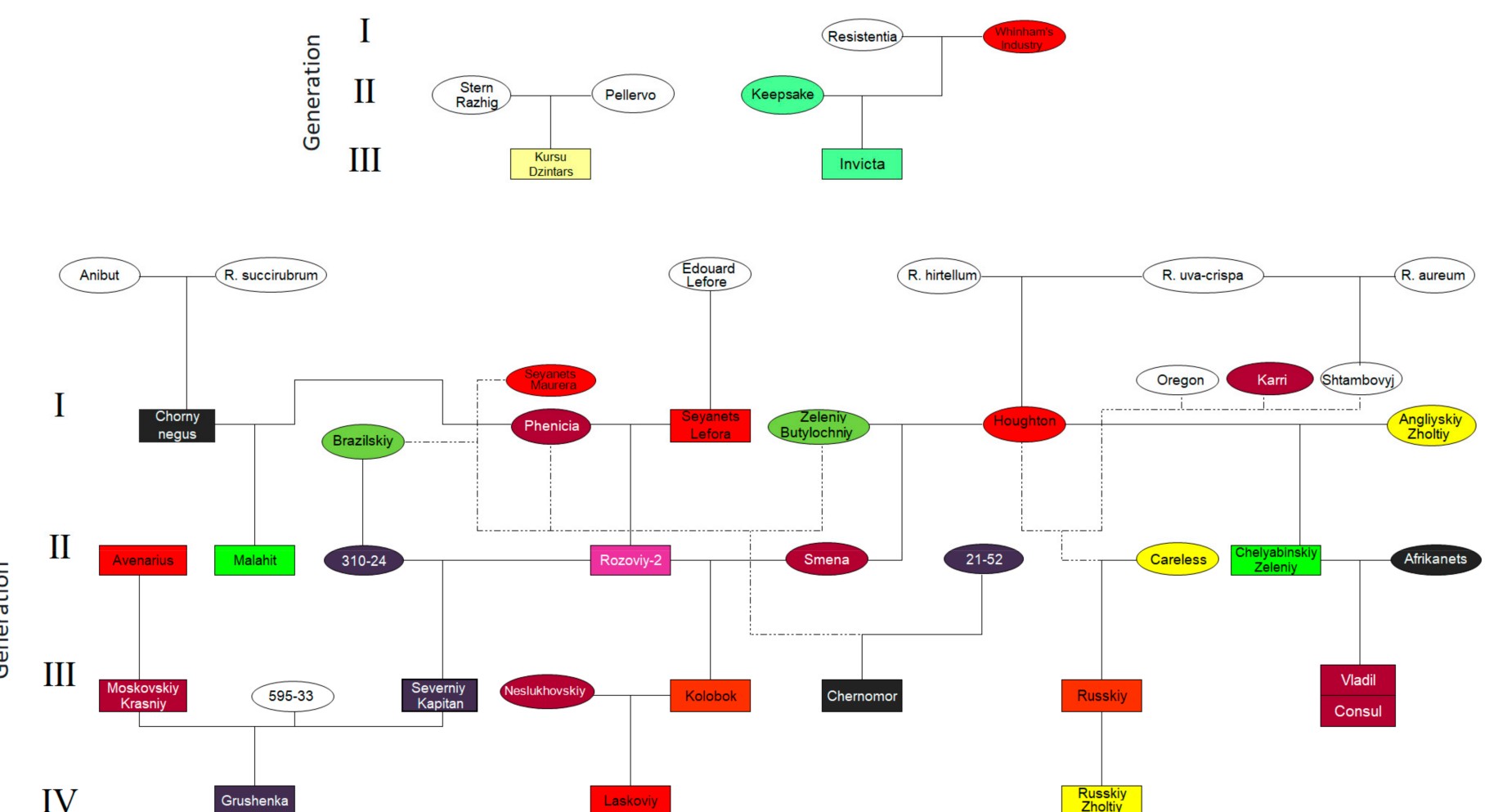

**Figure 1.** The genetic origin of cultivars under study (the studied cultivars are in quadrangles; connected by a dotted line are the cultivars whose mixed pollen was used for pollination).

**Table 2.** Description of 21 SSR loci in flavonoid biosynthesis genes and their polymerase chain reaction (PCR) primer pairs used to study genetic diversity in *Ribes uva-crispa* cultivars.

| Locus | Gene | Species | NCBI GenBank Accession Number | Motif and Number of Repeats | Location in the Gene | PCR Primer Nucleotide Sequence | | Allele Size, bp | |
|---|---|---|---|---|---|---|---|---|---|
| | | | | | | Forward | Reverse | Expected | Observed |
| RucANS | anthocyanidin synthase (*ANS*) | *Ribes uva-crispa* | LN736353 | (TA)$_6$ (AGTGA)$_2$ | Intron/exon | TCTTAACCCTAAAATTGCAGCC | CCATTCCACCAACTTCTTTCTC | 236 | 237, 241 |
| RucHLH-1 | *bHLH3* | *Ribes uva-crispa* | LN736347 | (ACTA)$_3$ | Intron | TTTCACTAGAGCCATTCTTGCC | GAAAATACGTTCACGATGGAGC | 208 | 198, 208 |
| RucHLH-2 | *bHLH3* | *Ribes uva-crispa* | LN736347 | (T)$_{10}$ | Intron | TTTTCTCTTCCTCGTGTTGCTC | CCCTCTCTGTAGTGCCAAATTC | 245 | 243, 247, 248 |
| RucHLH-3 | *bHLH3* | *Ribes uva-crispa* | LN736347 | (TTTCTC)$_2$ | Intron | GAATTTGGCACTACAGAGAGGG | TGAAGTTGAGTGTTCGGAGAGA | 317 | 314, 318, 319 |
| RucHLH-4 | *bHLH3* | *Ribes uva-crispa* | LN736347 | (AGA)$_5$ (GAG)$_4$ | Exon | CGTAAACCCTAACCGAGTCATC | ATTATTTGAAGCGTCGTCGG | 247 | 246 |
| RucWD-1 | *WD40* | *Ribes uva-crispa* | LN736318 | (CCA)$_4$ | Exon | CTTCTCGCCTACACCATCAAG | AAGAGTTGGGTTGGAACATGAG | 191 | 189 |
| RucWD-2 | *WD40* | *Ribes uva-crispa* | LN736318 | (CCAAC)$_2$ | Exon | CGACGAAACCCTAAGCATAAAG | CAAGCAATGTCGTAAACCTCCT | 372 | 378 |
| RucWD-3 | *WD40* | *Ribes uva-crispa* | LN736318 | (TTTGA)$_2$ | Exon | CCACGATAAGGAGGTTTACGAC | CCATCAAAATAGTCGCCATGTA | 203 | 204 |
| RucDFR1-1 | dihydroflavonol 4-reductase (*DFR*) | *Ribes uva-crispa* | LN736355 | (AAGAA)$_2$ (AAAC)$_3$ | Intron | AATAGGACGGAGGGAGTACGA | ACTTCATTCTGCCAAGTAGGGT | 396 | 399 |
| RucDFR1-2 | dihydroflavonol 4-reductase (*DFR*) | *Ribes uva-crispa* | LN736355 | (A)$_{15}$ (TTTTC)$_2$ | Intron | ACCCTACTTGGCAGAATGAAGT | CGTGGTCTTCGACACAAAATAC | 362 | 364, 370, 375, 377, 388, 390, 393, 397 |
| RucDFR1-3 | dihydroflavonol 4-reductase (*DFR*) | *Ribes uva-crispa* | LN736355 | (T)$_{19}$ | Intron | CTAGTGGTTGGTCCTTTCATCA | CTAGGCTGGTCCCTAAATCGTA | 329 | 318, 320, 323, 324, 326 |
| RucDFR1-4 | dihydroflavonol 4-reductase (*DFR*) | *Ribes uva-crispa* | LN736355 | (ATGTT)$_2$ | Intron | CGTGTATGAGAACCCTAAAGCC | CTTTTCGTCAATGCCCTTAAAC | 323 | 326 |
| RucDFR2-1 | dihydroflavonol 4-reductase (*DFR*) | *Ribes uva-crispa* | LN736356 | (TTAATT)$_2$ (A)$_{14}$ | Intron | CTATATCGTTCGAGCAACCGTA | TGGCAAGTCTAACAAATGCTTC | 259 | 257, 258, 259, 261 |
| RucUFGT | UDP glucose flavonoid 3-O-glycosyltransferase-like protein (*UFGT*) | *Ribes uva-crispa* | LN736351 | (TTGGAG)$_2$ (TTTAT)$_2$ (GCATA)$_2$ (TATGC)$_2$ (TTA)$_4$ (TATAG)$_2$ | Exon/Intron | GTGCTCATGTTTATACCGACTTCA | CAAAGCAAAGGGAAGAGGTTG | 351 | 343, 346, 355 |
| RucCHS-1 | chalcone synthase (*CHS*) | *Ribes uva-crispa* | LN736358 | (GCAAC)$_2$ | Exon | CACGAATCCACTTGTGTGTTTT | ATTTAGGTTGACCCCATTCCTT | 343 | 343 |
| RucCHS-2 | chalcone synthase (*CHS*) | *Ribes uva-crispa* | LN736358 | (CACGT)$_2$ | Exon | GAAGTGGGCCTTACATTTCATC | CTCGTCCAAGATAAACAACACG | 267 | 268 |
| RnMYB-1 | transcription factor *MYB 10* | *Ribes nigrum* | LN736314 | (TA)$_{11}$ | Intron | AGTGGTTTCGGAGTTAGGA | TCCAGGAATTCTTCCAGCAA | 340–350 | - |
| LfMYB-1 | *MYB113* | *Liquidambar formosana* | KU987934 | (AAATA)2 (AAAATA)1,8 (AAAAATA)2 (AAATAG)1,6 | Intron | ATTAGGAATGGAGGAGGCTAGG | CATGAGCTGGTTCACTTGACAT | 353 | - |
| LfMYB-2 | *MYB113* | *Liquidambar formosana* | KU987934 | (AGTGAA)2 (GCCTCG)2 | Exon | GAAAGATAGTGCCCAGAAGACG | TCTAAGGGAAACTCATTCCAGC | 375 | - |
| RJL-2 | microsatellite DNA, clone RJL-2 | *Ribes nigrum* | AJ439089 | (AG)$_{11}$ | - | CGAAGGTTGAATCGGTGAGT | TGTGGAACTACGTGGCTACG | 207 | 222, 228, 231, 234, 236, 240, 243, 254 |
| RJL-6 | microsatellite DNA, clone RJL-6 | *Ribes nigrum* | AJ439093 | (CT)$_{12}$(TTCA)$_3$(CT)$_6$ | - | TGTTCCCTGTTTCCTTCAAAA | GGACGTGGACGATGAGAGTT | 291 | 283, 285, 287, 291, 293, 295, 301 |

*2.3. DNA Extraction, PCR Conditions, and Fragment Analysis*

Total DNA was isolated from gooseberry leaves by a modified CTAB method [36] with an extraction buffer of the following composition: 2% CTAB (*w/v*), 100 mM Tris (pH 8.0), 20 mM EDTA (pH 8.0), 1.4 M NaCl. The extraction time was increased to 3 h. The DNA quality was tested by electrophoresis in 1% agarose gel. The DNA was quantified with a NanoDrop 2000 spectrophotometer (Thermo Fisher Scientific Inc., Waltham, MA, USA). The initial concentration of DNA in the samples was adjusted to 50 ng/μL in TE buffer. Each locus was genotyped using a matched primer pair composed of a forward primer labeled with 6-carboxyfluorescein (6-FAM) and a non-labeled reverse primer (Syntol Comp., Moscow, Russia). Amplification was performed in 25 μL of a mixture consisting of genomic DNA, 50 ng, the primers, 7 pmol each, and reagents from the EncycloPlus PCR kit (Evrogen JSC, Moscow, Russia). Amplification was performed on an MJ Mini thermal cycler (Bio-Rad Laboratories, Inc., Hercules, CA, USA). The procedure included initial denaturation at 95 °C for 3 min, followed by 34 cycles consisting of: 30-s denaturation at 95 °C, primer annealing at 60 °C (the optimal annealing temperature for all primer pairs) for 20 s, elongation at 72 °C for 40 s. The final elongation was performed at 72 °C for 10 min. For fragment analysis, we obtained stable and bright DNA fragments of the expected size for each locus (200–400 bp). If three amplifications had failed to generate a PCR product, a null allele was recorded for the genotype. Fragment analysis was performed by capillary electrophoresis on an ABI 3130xl Genetic Analyzer (Applied Biosystems, Foster, CA, USA). The S450 LIZ size standard (Syntol Comp., Moscow, Russia) was used as marker fragments. Peak identification and fragment sizing were performed using GeneMapper® software v. 4.0 (Applied Biosystems, Foster, CA, USA).

*2.4. Genetic Data Analysis*

Genetic statistics were calculated for each polymorphic marker. The observed (Ho) and expected (He) heterozygosities, the number of alleles, and the PIC (polymorphic information content) values were calculated using the Power Marker 3.25 software [37]. A UPGMA dendrogram was constructed using the MEGA X software package [38]. Principal component analysis (PCA) and construction of the box plots were performed with the PC–ORD 5 software [39].

## 3. Results

*3.1. Polymorphism and Genetic Diversity Analysis*

Nineteen microsatellite markers developed based on the sequences of structural and regulatory genes of flavonoid biosynthesis in *Ribes uva-crispa*, and two neutral markers (RJL-2 and RJL-6) were used to evaluate the genetic diversity of 22 gooseberry cultivars. Each PCR-generated sample consisted of one or two alleles. A total of 53 alleles were found as a result of the analysis. The number of alleles per locus varied from 2 (RucANS, RucCHS-1, RucCHS-2, RucDFR1-1, RucDFR1-4, RucHLH-1, RucHLH-4, RucWD-1, RucWD-2, RucWD-3) to 8 (RucDFR1-2, RJL-2). The mean number of alleles was 4.8 per locus. Of our developed markers and the neutral SSR markers, nine were polymorphic and nine monomorphic (Table 3). Two markers developed for the early flavonoid biosynthesis gene *CHS* were monomorphic. Three of five markers developed based on the late flavonoid biosynthesis gene *DFR* were polymorphic. All markers developed for the late biosynthesis genes *ANS* and UFGT were polymorphic. For the MBW complex genes, four markers were developed for *bHLH3* and three markers for each of *WD40* and *MYB* For the *bHLH3* gene, two markers proved to be polymorphic. All markers developed for *WD40* were monomorphic. Based on the known nucleotide sequences of MYB-family transcription factors of closely related species, three polymorphic markers were developed: one for the *MYB10* sequence (*Ribes nigrum*) and two for the *MYB113* sequence (*Liquidambar formosana*). Unfortunately, all the three markers developed for the MYB genes were not suitable for the gooseberry cultivars under study. The primers did not anneal or annealed non-specifically.

**Table 3.** Parameters of genetic variation for 9 polymorphic SSR loci in 22 gooseberry cultivars.

| Locus | Location in the Gene | Major Allele Frequency | Number of Alleles | Heterozygosity | | Polymorphism Information Content (PIC) |
|---|---|---|---|---|---|---|
| | | | | Expected ($H_e$) | Observed ($H_o$) | |
| RucDFR2-1 | intron | 0.39 | 4 | 0.71 | 0.95 | 0.66 |
| RucDFR1-2 | intron | 0.45 | 8 | 0.69 | 0.91 | 0.65 |
| RucDFR1-3 | intron | 0.48 | 5 | 0.67 | 0.77 | 0.63 |
| RucANS | intron/exon | 0.98 | 2 | 0.04 | 0.05 | 0.04 |
| RucUFGT | intron/exon | 0.93 | 3 | 0.13 | 0.14 | 0.12 |
| RucHLH2 | intron | 0.80 | 3 | 0.34 | 0.41 | 0.31 |
| RucHLH3 | intron | 0.64 | 3 | 0.52 | 0.73 | 0.47 |
| RJL-2 | - | 0.48 | 8 | 0.69 | 0.86 | 0.65 |
| RJL-6 | - | 0.39 | 7 | 0.74 | 1.00 | 0.71 |
| Mean | - | 0.61 | 4.8 | 0.51 | 0.65 | 0.47 |

The most polymorphic marker for the biosynthesis genes was RucDFR2-1, with a PIC value of 0.66 (Table 3). Loci residing within introns had the highest PIC values. Loci in the exon or intron-exon regions (Table 2) were monomorphic or less polymorphic and had a low PIC value.

Three unique alleles were identified: allele 364 in the locus RucDFR1-2, 323 in RucDFR1-3, and 257 in RucDFR2-1. They were found in Russian cultivar only. Foreign-bred cultivars (Invicta, Kursu Dzintars, Lepaan Red) and their direct derivatives (Chorny Negus, Seyanets Lefora) lacked those alleles. At all loci but RucANS, the expected heterozygosity (He) was lower than the observed one (Ho). SSRs located in the intron regions had a higher degree of genetic variation (expected and observed heterozygosity of 0.59 and 0.75, respectively) than SSRs in the intron-exon regions (expected and observed heterozygosity of 0.09 and 0.09, respectively). The highest PIC value was 0.71 for the neutral marker RJL-6. At the same time, high PIC values were also typical for all DFR loci, comparable to the PICs of neutral markers.

Ten of 53 identified alleles were unique (Table 4). It was noted that the number of unique alleles in cultivars with green or yellow berries was greater than in cultivars with red or dark red berries.

**Table 4.** Cultivar-specific unique alleles for SSR loci.

| Cultivar Name | Allele Size, bp | Locus |
|---|---|---|
| Angliyskiy Zeleniy | 393 | RucDFR1-2 |
| Angliyskiy Zeleniy | 234 | RJL-2 |
| Angliyskiy Zeleniy | 287 | RJL-6 |
| Malahit | 240 | RJL-2 |
| Nezhniy | 241 | RucANS |
| Nezhniy | 355 | RusUFGT |
| Nezhniy | 231 | RJL-2 |
| Kolobok | 222 | RJL-2 |
| Seyanets Lefora | 388 | RucDFR1-2 |
| Triumfalniy | 397 | RucDFR1-2 |

Common unique alleles were found in the British cultivar Invicta and the Finnish cultivar Lepaan Red: allele 324 at locus RucDFR1-3, allele 236 at locus RJL-2. The related cultivars Chorny Negus and Malahit also had a common unique allele: 346 at the locus RucUFGT. The cultivars Avenarius and Chorny Negus shared unique allele 390 for the RucDFR 1-2 marker. The cultivars Grushenka and Laskoviy had a large number of common unique alleles (similar in the degree of hardiness rating and thorniness): 375 for the locus RucDFR1-2, 326 for the locus RucDFR1-3, 314 for the locus RucDFR1-3, and 283 for the locus RJL-6. In addition, for all cultivars with a low degree of spinosity, the presence of

common and unique alleles 222, 228, 231, and 240 for the RJL-2 locus, 248 for the locus RucHLH2, and 314 for the locus RucHLH3 were noted (Supplemental materials: Table S1).

## 3.2. Cluster Analysis

The cultivar distribution in the dendrogram better correlates with geographical and genetic origin. In addition, a correlation has been noted between genetic origin and the degree of spinosity. A clear correlation between the genetic profile and winter hardiness, also the berry color was not found. The dendrogram is divided into three distinct clusters. The European-bred cultivars and their direct derivatives (Invicta, Kursu Dzintars, Lepaan Red, Chorny Negus, Seyanets Lefora) make up a separate group and differ significantly in their genetic profile from cultivars bred in Russia (Figure 2). A special position is held by Chorny Negus, an interspecies hybrid. It clearly differs from all other cultivars bred in Russia or in foreign countries. Russian cultivars form one big cluster. The green-berry cultivar Malahit falls into the group of cultivars with red-to-dark-red and black berries, probably due to its parental cultivars with dark-colored berries (Chorny Negus and Finicia). The analysis showed the genetic similarity of Rozoviy-2, Russkiy, and Russkiy Zholtiy. These varieties also have an identical degree of frost resistance and spinosity. Russkiy Zholtiy was derived by a bud mutation in plants of the Russkiy cultivar. A separate group includes green and yellow berry cultivars (Triumphalniy, Angliyskiy Zeleniy, Malahit, and Nezhniy) carrying unique alleles that may be specific to light-colored cultivars only. These varieties are characterized by a different degree of spinosity of the shoots but a similar high frost resistance. The varieties Grushenka and Laskoviy have the same genetic profile and the same degree of frost resistance. Moreover, both varieties are characterized by an almost complete absence of thorns. In addition, similar genetic profiles and degree of thorniness are varieties Vladil—Severniy Kapitan, Consul—Chernomor, and Avenarius and Triumfalniy.

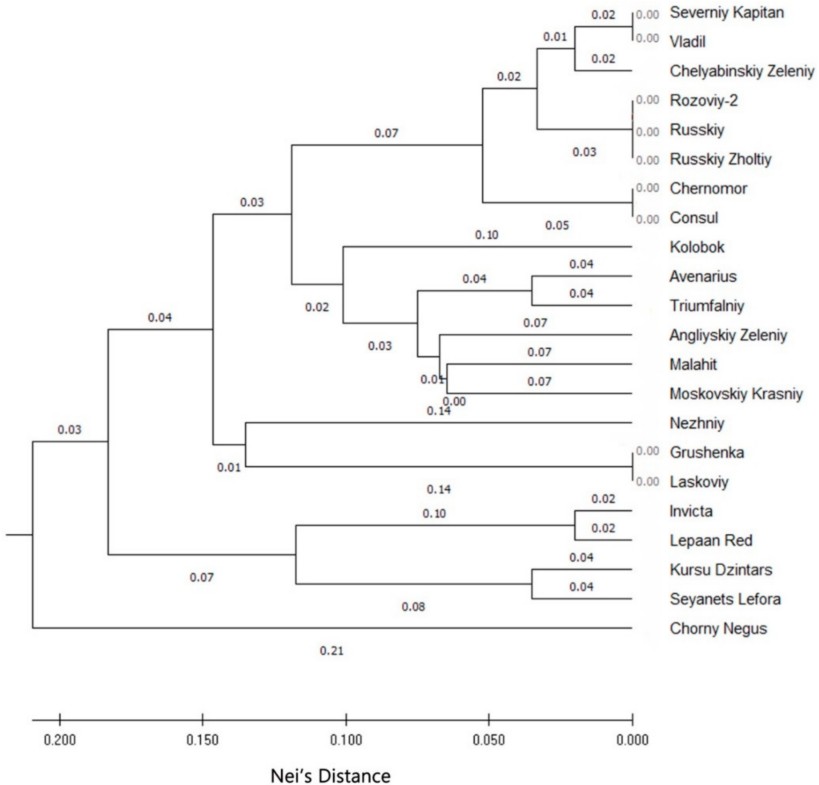

**Figure 2.** UPGMA dendrogram of 22 gooseberry cultivars based on pairwise Nei's standard genetic distances calculated using 7 SSR markers located in the flavonoid biosynthesis genes.

As shown by PCA analysis of the obtained data for polymorphic SSR markers, the cultivars under study can be divided into four groups (Figure 3) based on common genetic origin. One group includes the cultivars of European origin: Invicta, Kursu Dzintars, Lepaan Red, and Seyanets Lefora. Another group consists of cultivars derived by interspecies breeding and daughter cultivars (Chorny Negus and Malahit), as well as Russian cultivars of unknown origin (Avenarius and Triumfalniy). Cultivars of Russian origin make up one big group and are highly similar among themselves. The cultivar Nezhny makes a separate group on its own. Its origin probably differs from that of other cultivars under study. At the same time, its *DFR* gene loci have unique alleles characteristic for Russian cultivars only; it also has three unique alleles in the genes *ANS*, *UFGT* and in the neutral marker RJL-2 (alleles 241, 355, and 231, respectively). This cultivar is likely the result of a cross between a Russian cultivar and some foreign cultivar, genetically distant from the cultivars under study. In addition, five close subgroups were identified, formed by cultivars with common phenotypic characteristics associated with the degree of spinosity (Figure 3)

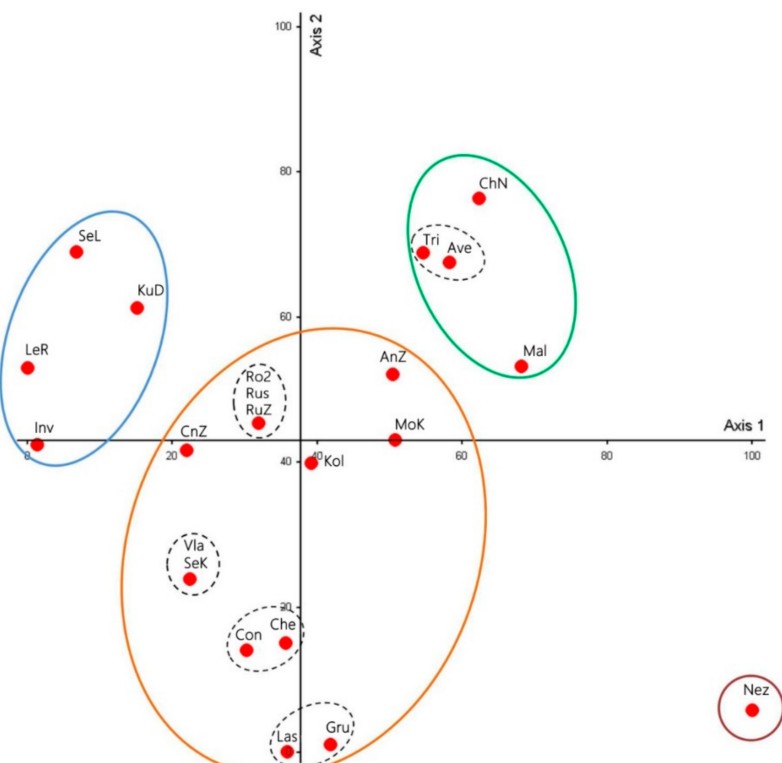

**Figure 3.** PCA of genotyped of 22 gooseberry cultivars based on 9 polymorphic SSR markers located in the flavonoid biosynthesis genes and neutral loci. The solid line marks groups of cultivars that have a close genetic origin; the dotted line marks subgroups with similar phenotypic characteristics.

## 4. Discussion

In recent years, the use of microsatellite markers (SSRs) in plants has proved to be a promising tool for genetic diversity analysis and mapping, cultivar and genotype certification, MAS, and identification of quantitative trait loci (QTLs) [40]. SSR markers were successfully used for studying the genetic diversity of the genus *Ribes* [28,33], establishing the degree of relatedness [33,41], and developing QTL-linked markers [7,10,42]. In those studies, however, genotyping was either performed for a few gooseberry cultivars or was not performed at all. We evaluated 22 gooseberry cultivars with different degrees of spinosity, frost resistance, and berry color and used 21 SSR loci. The developed loci resided in the known sequences of flavonoid biosynthesis pathway genes and in the regulatory factors involved in the regulation of flavonoid biosynthesis [26], affecting the formation of spines [27,43] and in the neutral areas not associated with any trait [34]. The biosynthesis

of flavonols and anthocyanins can affect both winter hardiness [20,44] and a change in the berries' color [25]. At the same time, plant spinosity is determined by the complex action of a large number of genes [45]. Therefore, variable and neutral markers with a high degree of probability can show differences in plants with different types of spinosity. It was also noted that the degree of plant spinosity could affect the frost resistance of plants [45].

The anthocyanin synthesis pathways of *Ribes* plants suggest the absence of the pelargonidin branch, which is a subset of the general flavonoid pathway [46]. Chalcone synthase (CHS) is one of the enzymes involved in the early flavonoid biosynthesis pathway. The pathway also includes chalcone isomerase (CHI) and flavanone 3-hydroxylase (F3H). The "late" pathway is mediated by dihydroflavonol 4-reductase (DFR), anthocyanin synthase (ANS), UDP glucose: flavonoid-3-O-glucosyltransferase (UFGT). The 'late' biosynthesis genes are regulated by the MBW complex, which includes the transcription factors MYB, bHLH, and WD40 [47].

Nine of the developed microsatellite loci were located in structural genes (RucCHS-1, RucCHS-2, RucDFR1-1, RucDFR1-2, RucDFR1-3, RucDFR1-4, RucDFR2-1, RucANS, RucUFGT), 10 in regulatory genes (RucHLH-1, RucHLH-2, RucHLH-3, RucHLH-4, RucWD-1, RucWD-2, RucWD-3, RnMYB-1, LfMYB-1, LfMYB-2), and two loci were neutral (RJL-2, RJL-6). Only 9 of all 21 studied marker sequences were found to be polymorphic. The mean number of alleles per polymorphic SSR locus in the flavonoid biosynthesis genes was 4.8, the mean values of He and Ho were 0.51 and 0.65, respectively, and the mean value of PIC was 0.47. These data are comparable to the earlier published mean values He = 0.6, Ho = 0.6, PIC = 0.55 for different currant cultivars and species [48]. Such similarities of our results and the above-mentioned study of European currant cultivars were probably due to the fact that both studies mostly used local cultivars.

Markers located in the *DFR* gene (RucDFR1-2, RucDFR1-3 and RucDFR2-1) are the most polymorphic among the developed markers of the flavonoid pathway genes. Their PICs (mean $PIC_{RucDFR}$ = 0.65) were significantly higher than those of other polymorphic markers (Table 3). The enzyme DFR comes into play in the mid-part of the late flavonoid biosynthesis pathway and is a key enzyme that ultimately controls the further synthesis pathway [49]. Based on their functional features, the *DFR* genes were proved to have higher variability than other structural genes of the flavonoid pathway [50]. This is also consistent with our findings. The high variability of the *DFR* gene was also confirmed by SSR analysis data for different raspberry cultivars. Developed for the *DFR* genes, two markers from *Rubus hybrid cultivar* and one marker from the closely related *Fragaria × ananassa* did not work on the studied raspberry and blackberry cultivars [51].

In addition to the regulation of the flavonoid pathway, genes of the regulatory complex MBW are also responsible for a number of functions in the plant organism [52], for example, such as the formation of spines analyzed by us [27]. Therefore, the MBW genes, especially *bHLH* and *MYB*, are highly variable [47]. We supposed that the marker sequences of these genes in the gooseberry cultivars under study would be polymorphic. We used loci in the sequences of the transcription factors *WD40, bHLH3, MYB10*, and *MYB113*. Unfortunately, all three markers selected for the MYB genes did not suit the studied gooseberry cultivars. That was likely due to the fact that in the absence of known MYB gene sequences for gooseberry, we had to construct the primers based on the MYB sequences of related plants, such as *Liquidambar formosana* and *Ribes nigrum*. Probably, the high variability of this transcription factor [25,47] prevented us from identifying common loci even in the related plant species. Our data are consistent with those of Starkevič and co-authors (2020), who showed *WD40* to be the most conservative transcription factor in the MBW complex [25]. All three developed markers proved to be monomorphic in all cultivars under study. For the gene *bHLH3*, two studied loci were monomorphic (RucHLH-1, RucHLH-4), and two were polymorphic (RucHLH-2, RucHLH-*3*). The high variability of the *bHLH* genes [53] was also supported by our results.

In addition, to assess the genetic variability of different popular gooseberry cultivars grown in the Russian Federation and for cultivars, certifications were analyzed using

the neutral markers RJL-2 and RJL-6 tested by Brennan R et al. on representatives of the genus *Ribes* [34]. These loci had also been tested on currant and gooseberry by other researchers [12,32]. The markers are polymorphic, with the number of alleles reaching 16 (RJL-2) or 13 (RJL-6) in different *Ribes* species [34]. We found 8 (RJL-2) and 7(RJL-6) alleles for the marker sequences for the gooseberry cultivars under study. In our study, these loci amplified in all gooseberry cultivars and showed a high level of polymorphism ($PIC_{RJL-2}$ = 0.65, $PIC_{RJL-6}$ = 0.7) and heterozygosity ($Ho_{RJL-2}$ = 0.86, $Ho_{RJL-6}$ = 1.0) among both European and Russian cultivars. The size of alleles obtained for the gooseberry cultivars under study was within the previously described ranges for gooseberry cultivars of different genetic origins [32]. It is noteworthy that the PIC values of the neutral loci were comparable with those of the most polymorphic loci residing in the *DFR* gene ($PIC_{RucDFR2-1}$ = 0.66, $PIC_{RucDFR1-2}$ = 0.65, $PIC_{RucDFR1-3}$ = 0.63). Thus, the RucDFR loci can also be used as part of multiplex test systems for the genetic and phenotypic identification of gooseberry cultivars.

We have noted a direct relationship between the homogeneity of a motif and its repeat frequency: the more homogeneous the motif and the greater the number of repeats, the higher the polymorphism of the locus. The SSR regions of all highly polymorphic RucDFR loci are characterized by long mononucleotide repeats (Table 2). The locus RucHLH-2 was shown to have a long mononucleotide repeat in the SSR region, which correlates with the high PIC value for this locus ($PIC_{RucHLH-2}$ = 0.31). The polymorphic neutral loci contain long dinucleotide sequences. Similar studies have confirmed that it is the motif homogeneity and the repeat frequency that are directly related to the number of alleles and the degree of heterozygosity compared with more heterogeneous and less frequently repeated motifs [34,51,54]. Such motifs were more typical of introns. In our study, the highest polymorphism was noted for the intronic loci RucDFR2-1, RucDFR1-2, RucDFR1-3, RucHLH2, RucHLH3 (PICs ranged from 0.31 to 0.66). A lower polymorphism was observed for intron-exon regions of the RucANS and RucUFGT loci (PICs ranged from 0.04 to 0.12). Monomorphic loci were obtained for all studied exon regions. Our research supports other studies indicating that the higher variability of introns is due to a weaker pressure of natural selection [55].

The cluster and PCA analyses showed a clear division between the cultivars of European and hybrid origin and the Russian-bred cultivars (Figures 2 and 3). All Russian cultivars had unique alleles for all polymorphic loci of the *DFR* gene (RucDFR2-1—257 bp, RucDFR1-2—364 bp, RucDFR1-3—323 bp) and were therefore united in a separate cluster. Similar results were obtained in a genotyping study of genus *Ribes* cultivars grown in Belarus: Russian-bred cultivars (Severniy Kapitan and Malahit) also formed their own cluster, separate from the European-bred cultivars [33]. Therefore, it can be assumed that most Russian gooseberry cultivars share a common ancestor. A separate cluster was also formed by the interspecies hybrid Chorny Negus, its descendent cultivar Malahit and the two Russian cultivars of unknown origin, Avenarius and Triumfalniy. Malahit is somewhere in-between: according to the dendrogram, it is more closely related to the Russian cultivars because it has certain characteristic alleles in the *DFR* gene loci, yet PCA shows its greater proximity to Chorny Negus. The two cultivars were shown to have a unique allele 346 in the rather conservative locus RucUFGT, whereas other cultivars, including those bred in Europe, had only allele 343. Avenarius and Chorny Negus share unique allele 390 for the marker RucDFR1-2, which may indicate their kinship. Triumphalniy has a unique allele 397 for the marker RucDFR1-2, and the cultivar's overall allelic composition makes it genetically closer to the group of Chorny Negus, Avenarius, and Malahit. The cultivar Nezhniy holds a stand-alone position. In its *DFR* gene loci, Nezniy has alleles that are specific to the Russian cultivars. However, this cultivar was shown to have unique alleles in the loci of the *ANS* and *UGT* genes, as well as in the neutral marker (Table 4). Such a unique set of markers for the cultivar is probably due to its genetic origin from an ancestor shared with Russian cultivars and another ancestor genetically distant from the European cultivars under study. We were unable to show differences in allele size for the cultivar

Russkiy Zholtiy, which is a bud mutation, and its parent cultivar Russkiy. This means the presence of either a mutation in genes not used in our study or a single nucleotide replacement undetectable by fragment analysis. At the same time, the genetic profiles of cultivars of Russian breeding, even those of distant origin but having the same level of frost resistance, are similar in the studied varieties. Common alleles for these varieties were found in the *DFR* genes (RucDFR1-2—375, RucDFR1-3—326). It is described that a change in the biosynthesis of flavonoids can affect the degree of frost resistance for plants [20], which can explain our results.

The cultivar distribution within the main clusters depended more on their genetic origin and degree of spinosity than on the fruit color and degree of frost resistance. Cultivars with common ancestors, such as Houghton, were grouped together (Figure 1). However, because the cultivar selection was mainly performed among closely related varieties and they probably shared a common ancestor, these Russian-bred cultivars have a common pool of specific alleles in genes such as *DFR*. Our study has failed to find for the studied gooseberry cultivars any suitable marker sequences, which would have helped establish the linkage between the genetic component and the fruit color in gooseberry plants. That was probably because the analysis did not include all flavonoid biosynthesis genes due to the lack of known sequences of some of them. We have also failed to obtain amplifiable marker sequences for the most variable gene *MYB*. As shown by other researchers, some variations in an *MYB* gene sequence may be crucial for the fruit color in the genus *Ribes* [25] as well as in other plant species [56]. However, we have noted that cultivars with light-colored berries have a greater number of unique alleles than those with red and dark-colored berries, and they also form a small group within a common cluster. The cultivars with a low degree of spinosity were characterized by the presence of unique alleles in the *bHLH* gene and the RJL-2 neutral marker (RucHLH2—248, RucHLH3—314, RJL-2—222, 228, 231, 240). This is probably due to the fact that genes of the MBW complex, in particular *bHLH*, play a key role in the formation of spines at the first stages [27]. However, due to the fact that plant thorniness is determined by the complex action of a large number of genes [45], we can observe the emergence of unique alleles in neutral markers, which are the most variable.

## 5. Conclusions

Our study has demonstrated that the developed set of gene-targeted SSR markers representing structural and regulatory genes of the flavonoid biosynthesis pathway can be used for assessing genetic variability and relatedness among gooseberry cultivars. This study can be considered as laying the basis for marker-assisted selection of gooseberry plants with certain traits. In the course of the research, we have identified *DFR* gene alleles that are specific to Russian-bred gooseberry cultivars and suggest the presence of a common ancestor. The polymorphic SSR markers we have developed for the *DFR* gene are comparable in variability with neutral SSR markers. Therefore, these highly polymorphic markers can be recommended for use in multiplex systems for genetic identification and certification of gooseberry cultivars. The dependence of the low degree of the spine and the presence of common and unique alleles in the *bHLH* gene and the neutral marker RJL-2 were found. No direct correlation was found between the degree of frost resistance and the genetic component; however, unrelated varieties with similar frost resistance were genetically close in the *DFR* gene. In addition, we have not found a clear correlation between the genetic variations of the used flavonoid biosynthesis markers and the fruit color. Cultivars with light-colored berries, however, had more unique alleles than cultivars with red and dark-colored berries. Establishing a clearer relationship between the genetic variation of flavonoid biosynthesis genes and the fruit color requires having the full-length sequences of other gooseberry flavonoid biosynthesis genes.

**Supplementary Materials:** The following are available online at https://www.mdpi.com/article/10.3390/agronomy11061050/s1, Table S1: Cultivar-specific alleles for SSR loci of the analyzed gooseberries.



**Author Contributions:** Conceptualization, E.O.V. and K.A.S.; methodology, V.G.L., E.O.V., and N.M.S.; software, T.N.L.; validation, V.G.L., E.O.V., and N.M.S.; formal analysis, V.G.L. and K.A.S.; investigation, E.O.V.; resources, E.I.T.; data curation, E.O.V. and N.M.S.; writing—original draft preparation, E.O.V.; writing—review and editing, E.O.V., V.G.L., K.A.S., and N.M.S.; visualization, N.M.S. and T.N.L.; supervision, K.A.S.; project administration, K.A.S.; funding acquisition, E.I.T. and K.A.S. All authors have read and agreed to the published version of the manuscript.

**Funding:** This research was carried out within the state program of the Ministry of Science and High Education of the Russian Federation (theme "Plant molecular biology and biotechnology: their cultivation, pathogen and stress protection (BIBCH)" (No. 0101-2019-0037).

**Institutional Review Board Statement:** Not applicable.

**Informed Consent Statement:** Informed consent was obtained from all subjects involved in the study.

**Data Availability Statement:** Data are available upon request to the authors.

**Conflicts of Interest:** The authors declare no conflict of interest.

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
