# Peer review of "The Development of the Genic SSR Markers for Analysis of Genetic Diversity in Gooseberry Cultivars"

_agronomy, doi:10.3390/agronomy11061050_

Round 1

Reviewer 1 Report

In the current study, the authors have developed numerous SSR markers located in the flavonoid biosynthetic pathway genes used for genetic diversity analysis in gooseberry cultivars. In short, three unique alleles were identified that is the main novelty of this work. To me, the authors have done a great job and presented good data. Overall, there is no issue with the experimentation, and this manuscript can be considered for publication in Agronomy.

  • Line 56, please do not shuffle the order of QTL abbreviations. It should be Quantitative trait loci (QTL).

Author Response

Dear Reviewer,

Point 1:

ʺ Line 56, please do not shuffle the order of QTL abbreviations. It should be Quantitative trait loci (QTL).ʺ

Response 1:

Thank you for handling and thoroughly reviewing our manuscript. Corresponding corrections have been made (lines 55).

Best regards, on behalf of all authors,

- Konstantin

Reviewer 2 Report

The study described in this manuscript is a genetic characterization of gooseberry cultivars. In general, it is well-written, the methods are accurate, and the results are well described and interpreted. Unfortunately, the authors did not achieve the aim of the study, that was generate markers associated with flavonoid content. I consider that the authors could focus more on the description of the flavonoid biosynthetic pathway in the introduction, maybe using a schematic to ease understanding. Also, since they cannot find any association between the phenotype and the markers, at least they should discuss it further in the discussion: maybe these genes are not the most important ones in gooseberry, other regulators could exist, or other enzymes. It may be the low density of markers, or the low number of genotypes analyzed... I do consider that negative results are also results and the authors have done as much as possible with the study to present it in a nice way and take the most out of it.

Strengths:

The manuscript is easy to read, well written and has appropriate information.

The statistical analyses are correct and the authors have performed adequate analyses for the type of data generated.

It is interesting to focus genetic studies on understudied crops as gooseberry.

Limitations:

Limited number of cultivars (22) and markers (22).

No SSR for one of the genes - the ones they tried did not work and they did not design more primers to fix it.

Negative results, since the markers do not explain the fruit color.

Limited phenotyping - the authors only described the color visually but did not quantify the actual compounds causing those colors.

Find small details from the text below:

L84: In my opinion, currently the preferred markers are SNPs, SSR are useful and cheap to genotype but much more labor-intensive and nowadays many research groups have available SNP platforms or even sequencing platforms to genotype at medium-high scale more efficiently. I would ask the authors to downplay this statement and mention also other types of markers.
L110: Fig. 1 shows a genealogic tree/breeding scheme, not a phylogenetic tree.
L164, 179, 180: species without italics
L192-194: I am confused about this statement. Which are the heterozygosities for exonic and intronic regions? Please rephrase.
L205: Table 4 does not contain the size in bp of the alleles, as the header indicates.
L236: Figures do not have a color legend.

Author Response

Dear Reviewer,

Thank you for handling and thoroughly reviewing our manuscript. We appreciate the reviewers’ critique, recommendations, and comments that greatly helped us improve the manuscript. We took them all into consideration while revising the manuscript and provided below our detailed step-by-step explanation on how we addressed them. All changes are highlighted in yellow in the revised manuscript.

Point 1:

  1. ʺI consider that the authors could focus more on the description of the flavonoid biosynthetic pathway in the introduction, maybe using a schematic to ease understanding.ʺ

Response 1: A more complete description of the flavonoid biosynthetic pathway is presented from the Discussion (lines 257-264), where in our opinion it is more appropriate, including this information in the Introduction will cause a tautology. At the request of the reviewer, we can transfer this information to the Introduction.

Point 2:

  1. ʺAlso, since they cannot find any association between the phenotype and the markers, at least they should discuss it further in the discussion: maybe these genes are not the most important ones in gooseberry, other regulators could exist, or other enzymes. It may be the low density of markers, or the low number of genotypes analyzed...ʺ

Response 2: In the Discussion, information was added about unexplored genes that can probably affect the color change in gooseberries (lines 378-383).

Point 3:

  1. ʺ L84: In my opinion, currently the preferred markers are SNPs, SSR are useful and cheap to genotype but much more labor-intensive and nowadays many research groups have available SNP platforms or even sequencing platforms to genotype at medium-high scale more efficiently. I would ask the authors to downplay this statement and mention also other types of markers. ʺ

Response 3: The proposed changes have been made in the line 83.

Point 4:

  1. ʺ L110: Fig. 1 shows a genealogic tree/breeding scheme, not a phylogenetic tree. ʺ

Response 4:  In this case, the most appropriate expression is "pedigree reconstruction". Corresponding changes have been made in the line 109.

Point 5:

  1. ʺ L164, 179, 180: species without italicsʺ

Response 5: Corresponding corrections have been made (lines 162, 177, 178).

Point 6:

  1. ʺ L192-194: I am confused about this statement. Which are the heterozygosities for exonic and intronic regions? Please rephrase. ʺ

Response 6: The sentence has been rephrased and clarified (lines 190-193).

Point 7:

  1. ʺ L205: Table 4 does not contain the size in bp of the alleles, as the header indicates. ʺ

Response 7: Information was lost in the table, possibly when transferring from one file to another. The correct information has been added to Table 4.

Point 8:

  1. ʺ L236: Figures do not have a color legend. ʺ

Response 8: Provided full names in the Figure 2. The caption to the picture was expanded.

Best regards, on behalf of all authors,

- Konstantin

Round 2

Reviewer 2 Report

The authors have addressed the minor comments I provided and the manuscript is now correct and reflects the work done and the obtained results. However, no comment has been made about the limitations of the study.

Author Response

Dear Reviewer,

We are very grateful that you have carefully processed and reviewed our manuscript. We took into account the editor's wish to focus on the genetic diversity of the gooseberry cultivars under study. We have added new phenotypic data of the studied cultivars, which made it possible to focus more on molecular analysis to assess genetic diversity. This also made it possible to find new patterns between the genetic and phenotypic components. We have made all the required adjustments. In the modified manuscript, all changes are highlighted in light-blue.

Best regards, on behalf of all authors,

- Konstantin